# Ultrasensitive and visual detection of SARS-CoV-2 using all-in-one dual CRISPR-Cas12a assay

Xiong Ding [1], Kun Yin[1], Ziyue Li [1], Rajesh V. Lalla[2], Enrique Ballesteros[3], Maroun M. Sfeir[3] & Changchun Liu [1✉]

The recent outbreak of novel coronavirus (SARS-CoV-2) causing COVID-19 disease spreads rapidly in the world. Rapid and early detection of SARS-CoV-2 facilitates early intervention and prevents the disease spread. Here, we present an All-In-One Dual CRISPR-Cas12a (AIOD-CRISPR) assay for one-pot, ultrasensitive, and visual SARS-CoV-2 detection. By targeting SARS-CoV-2's nucleoprotein gene, two CRISPR RNAs without protospacer adjacent motif (PAM) site limitation are introduced to develop the AIOD-CRISPR assay and detect the nucleic acids with a sensitivity of few copies. We validate the assay by using COVID-19 clinical swab samples and obtain consistent results with RT-PCR assay. Furthermore, a low-cost hand warmer (~$0.3) is used as an incubator of the AIOD-CRISPR assay to detect clinical samples within 20 min, enabling an instrument-free, visual SARS-CoV-2 detection at the point of care. Thus, our method has the significant potential to provide a rapid, sensitive, one-pot point-of-care assay for SARS-CoV-2.

[1] Department of Biomedical Engineering, University of Connecticut Health Center, 263 Farmington Ave., Farmington, CT 06030, USA. [2] Section of Oral Medicine, University of Connecticut Health Center, Farmington, CT 06030, USA. [3] Department of Pathology and Laboratory Medicine, University of Connecticut Health Center, Farmington, CT 06030, USA. ✉email: chaliu@uchc.edu

Severe acute respiratory syndrome Coronavirus 2 (SARS-CoV-2, previously named 2019-nCoV) is a new coronavirus causing coronavirus disease 2019 (COVID-19) which first emerged in December 2019[1]. As of August 19, 2020, according to the World Health Organization (WHO)[2], 21,938,207 people all over the world have been infection-confirmed and 775,582 people have died. Rapid and early detection of this deadly virus plays a critical role in facilitating early intervention and treatment, which, in turn, may reduce disease transmission risk.

Polymerase chain reaction (PCR) method including its variant reverse transcription PCR (RT-PCR) is the most commonly used technology for pathogen nucleic acid detection and has been considered as a gold standard for infectious disease diagnostics due to its high sensitivity and specificity[3–5]. However, it typically relies on expensive equipment and well-trained personnel, as well as needs long assay reaction time (~2 h), all of which is not suitable for simple, rapid, and point of care (POC) molecular diagnostics of the SARS-CoV-2. In recent decades, several isothermal amplification methods, such as recombinase polymerase amplification (RPA)[6] and loop-mediated isothermal amplification (LAMP)[7], have been developed as attractive alternatives to conventional PCR method because of their simplicity, rapidity and low cost. However, there is still a challenge to apply them to develop accurate and reliable POC testing due to undesired nonspecific amplification signals which may cause false-positive results for SARS-CoV-2 assay[8,9].

Recently, RNA-guided CRISPR/Cas nuclease-based nucleic acid detection has shown great promise for the development of next-generation POC molecular diagnostics technology due to its high sensitivity, specificity, and reliability[10,11]. For example, several Cas nucleases, such as Cas12a, Cas12b, and Cas13a, perform strong collateral cleavage activities in which the Cas nucleases activated by CRISPR RNA (crRNA)-target duplex can indiscriminately cleave surrounding nontarget single-stranded nucleic acids[12–16]. By combining with RPA pre-amplification, Cas13 and Cas12a nucleases have, respectively, been used to develop SHERLOCK (Specific High-sensitivity Enzymatic Reporter UnLOCKing) system[17] and DETECTR (DNA Endonuclease-Targeted CRISPR Trans Reporter) system[13] for highly sensitive and specific nucleic acid detection. Apart from the RPA pre-amplification method, some CRISPR-Cas-based nucleic acid detection utilized LAMP and PCR pre-amplification, such as CRISPR-Cas12b-assisted HOLMESv2 platform and SARS-CoV-2 DETECTR[15,18]. However, these CRISPR-Cas-based nucleic acid detection methods typically require separate nucleic acid pre-amplification and multiple manual operations, which undoubtedly complicates the testing procedures and potentially increases the risk of carry-over contaminations due to amplification products transferring.

In this study, we report an all-in-one dual CRISPR-Cas12a (termed AIOD-CRISPR) assay for simple, rapid, ultrasensitive, specific, one-pot, and visual detection of SARS-CoV-2. Dual crRNAs without protospacer adjacent motif sites (PAM) sequence limitation are introduced to initiate dual CRISPR-based nucleic acid detection with high efficiency. In our AIOD-CRISPR assay, all components for nucleic acid amplification and CRISPR-based detection are thoroughly mixed in a single, one-pot reaction system, and incubated at a single temperature (e.g., 37 °C), eliminating the need for separate pre-amplification and transfer of amplified product. By targeting the nucleoprotein (N) gene of SARS-CoV-2, our AIOD-CRISPR assay method is able to detect few copies of the nucleic acids (DNA or RNA). In addition, our AIOD-CRISPR method has been validated by testing 28 clinical swab samples and obtained consistent results with that of RT-PCR method. Furthermore, a low-cost hand warmer has been directly used as its incubator for instrument-free POC diagnostics of COVID-19.

## Results

**AIOD-CRISPR assay system.** As shown in Fig. 1a, the AIOD-CRISPR assay system uses a pair of Cas12a-crRNA complexes generated by two individual crRNAs to bind two different sites which are close to the recognition sites of primers in the target sequence. The Cas12a-crRNA complexes are first prepared prior to being adding into the reaction solution containing RPA primers, single-stranded DNA fluorophore-quencher (ssDNA-FQ) reporter, recombinase, single-stranded DNA binding protein (SSB), strand-displacement DNA polymerase, and target sequences. When incubating the AIOD-CRISPR reaction system in one pot at ~37 °C, the RPA amplification is initiated and exposes the binding sites of the Cas12a-crRNA complexes due to the strand displacement. On one hand, when the Cas12a-crRNA complexes bind the target sites, the Cas12a endonuclease is activated and cleaves the ssDNA-FQ reporters, generating strong fluorescence signals. On the other hand, the amplified products generated during the RPA continuously trigger CRISPR-Cas12a-based collateral cleavage activity. Previous studies[13,16] have demonstrated that the collateral cleavage activity of the CRISPR-Cas12a system is independent of target strand cleavage. Therefore, target sequences for our AIOD-CRISPR assay are not limited by the Cas12a's PAM sequences[19].

To systematically evaluate our AIOD-CRISPR assay, we prepared and tested eight reaction systems (reactions # 1–8) with various components (Fig. 1b). A plasmid containing 316 bp SARS-CoV-2 N gene fragment, termed N plasmid, was used as the target sequence (Supplementary Fig. 1). The ssDNA-FQ reporter was a 5 nucleotide (nt) single-stranded DNA (5′-TTATT-3′) labeled by 5′ 6-FAM (Fluorescein) fluorophore and 3′ Iowa Black FQ quencher. After incubation at 37 °C for 40 min, only reaction # 4 containing target nucleic acid sequence, dual crRNAs, Cas12a, and RPA reaction mixture produced super-bright fluorescence signal (Fig. 1b), which could be directly visualized under a blue LED or UV light illuminator. Surprisingly, even under ambient light conditions without excitation, a color change from orange-yellow to green was directly observed in the reaction tube # 4 by naked eyes. To further verify the specificity of the generated fluorescence signal, the assay products (self-probed fluorescence reporters) were subjected to denaturing polyacrylamide gel electrophoresis (PAGE). As shown in Fig. 1b, a strong band with shorter DNA size was observed only in the lane of reaction # 4, which resulted from the cleaved ssDNA-FQ reporters with strong fluorescence signal. In comparison, for other reaction systems, only weak bands with relatively longer DNA sizes were observed in their corresponding lanes, which may be attributed to fluorescence quench of the intact uncut ssDNA-FQ reporters. In addition, in real-time fluorescence curves, only reaction # 4 showed a significantly increased fluorescence signal that saturated at 21 min (Fig. 1b). Thus, these results show that our AIOD-CRISPR assay provides a simple, rapid, one-pot approach for target-specific nucleic acid detection.

Since a previous study reported that RPA amplification reaction was initiated after adding MgOAc[20], we are interested in knowing if nucleic acid amplification is efficiently initiated at room temperature during sample preparation in our AIOD-CRISPR assay system. We prepared two AIOD-CRISPR solutions (one positive and one negative) and allowed them to remain at room temperature for 20 min. As shown in Supplementary Fig. 2, very weak fluorescence change between positive and negative samples was observed in the AIOD-CRISPR systems at room temperature. In comparison, there's a significant fluorescence change at 37 °C after as short as 10 min incubation (Supplementary Fig. 2). Eventually, the fluorescence signal was saturated and a color change from orange-yellow to green was present after 20-min incubation at 37 °C. Therefore, the AIOD-CRISPR assay

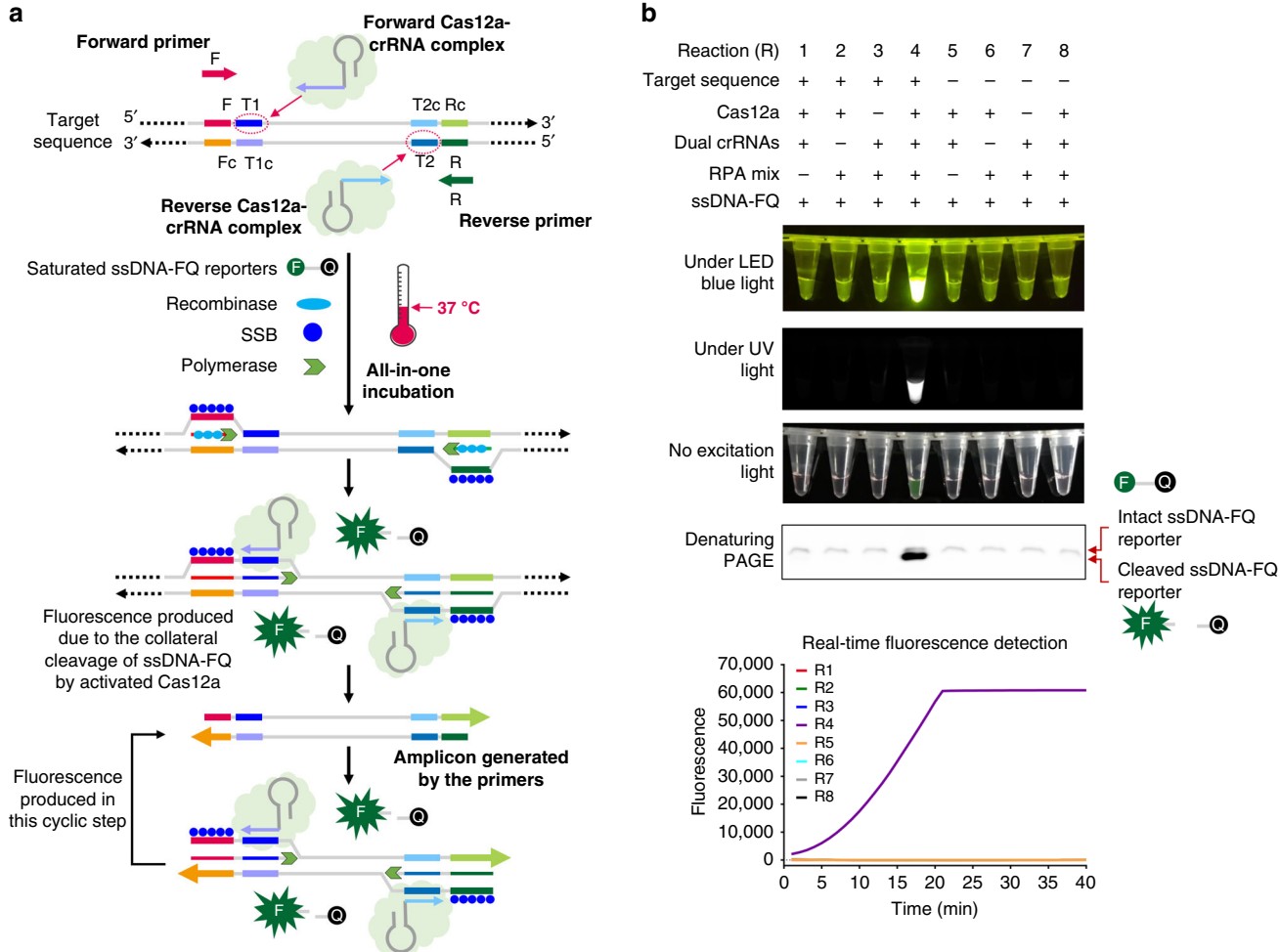

**Fig. 1 Design and working principle of the AIOD-CRISPR assay. a** Schematic of the AIOD-CRISPR assay system. SSB is single-stranded DNA binding protein. The four sites in the target sequence are labeled as F, T1, T2, and R, respectively. The letter c represents the corresponding complementary site. For example, F and Fc sites are complementary. The short horizontal lines with the same colors denote the same sites and their arrows represent the direction of 5′–3′. **b** Evaluation of eight AIOD-CRISPR reactions (R) with various components through endpoint imaging after 40-min incubation, denaturing polyacrylamide gel electrophoresis (PAGE) analysis of the single-stranded fluorescent reporter (ssDNA-FQ), and real-time fluorescence detection. The ssDNA-FQ was labeled by 5′ 6-FAM (Fluorescein) fluorophore and 3′ Iowa Black FQ quencher. Recombinase polymerase amplification (RPA) mix from TwistAmp Liquid Basic kit was composed of 1× Reaction Buffer, 1× Basic E-mix, 1× Core Reaction Buffer, 14 mM MgOAc, 320 nM each of primers, and 1.2 mM dNTPs. Dual crRNAs contained 0.64 µM each of crRNAs specific to the SARS-CoV-2 N gene sequence. A plasmid containing the SARS-CoV-2 N gene sequence ($3 \times 10^3$ copies), 8 µM of ssDNA-FQ reporters, and 1.28 µM EnGen Lba Cas12a (Cpf1) were used. Each experiment was repeated three times with similar results.

system is mainly triggered after reaction temperature is elevated to ~37 °C.

**Optimization of AIOD-CRISPR assay**. We first optimized ssDNA-FQ reporters in our AIOD-CRISPR assay because the reporter concentration plays a crucial role in fluorescence read-out. As shown in Supplementary Fig. 3a, b, the higher the concentration of the ssDNA-FQ reporters, the shorter the threshold time. As to fluorescence intensity, the minimal concentration for saturated values was 4 µM after 40 min incubation (Supplementary Fig. 3c). For better visual colorimetric detection, 8 µM ssDNA-FQ was the best choice (Supplementary Fig. 3d). Collateral cleavage efficiency of the activated Cas12a nuclease represents an ability to cut ssDNA-FQ reporters around it[12,13]. Thus, increasing the ssDNA-FQ reporter concentration can improve the fluorescence signals. By choosing 8 µM ssDNA-FQ, we next investigated the effect of the primer concentration and the ratio of crRNAs to Cas12a on the AIOD-CRISPR assay by

fixing the concentration of each crRNA at 0.64 µM. As shown in Supplementary Figs. 4 and 5, the optimal concentration of the primers and the optimal ratio of crRNAs to Cas12a was 0.32 µM and 1:1, respectively.

The primers are designed to amplify 121 bp of SARS-CoV-2 N gene sequence at the location from 28857 to 28977 in the SARS-CoV-2 genome map (GenBank accession MT688716.1) as shown in Fig. 2a. The four target sites for primers and crRNAs were highly conserved for SARS-CoV-2 with a theoretical mutation rate of <1.5%, according to the public data provided by Global Initiative on Sharing All Influenza Data (GISAID; Supplementary Fig. 6)[21]. Previous studies[12,13] have proved that the collateral cleavage of ssDNA-FQ reporters by the Cas12a nuclease is triggered by the binding of crRNA to target sites. Thus, the AIOD-CRISPR assays were investigated by using three different crRNAs (crRNA1, crRNA2, and crRNA3; Fig. 2a) which were specific to different sites in the same amplification region. Among them, crRNA3 was designed with PAM site (5′-TTTG-3′) limitation, while crRNA1 and crRNA2 not. Under the optimized

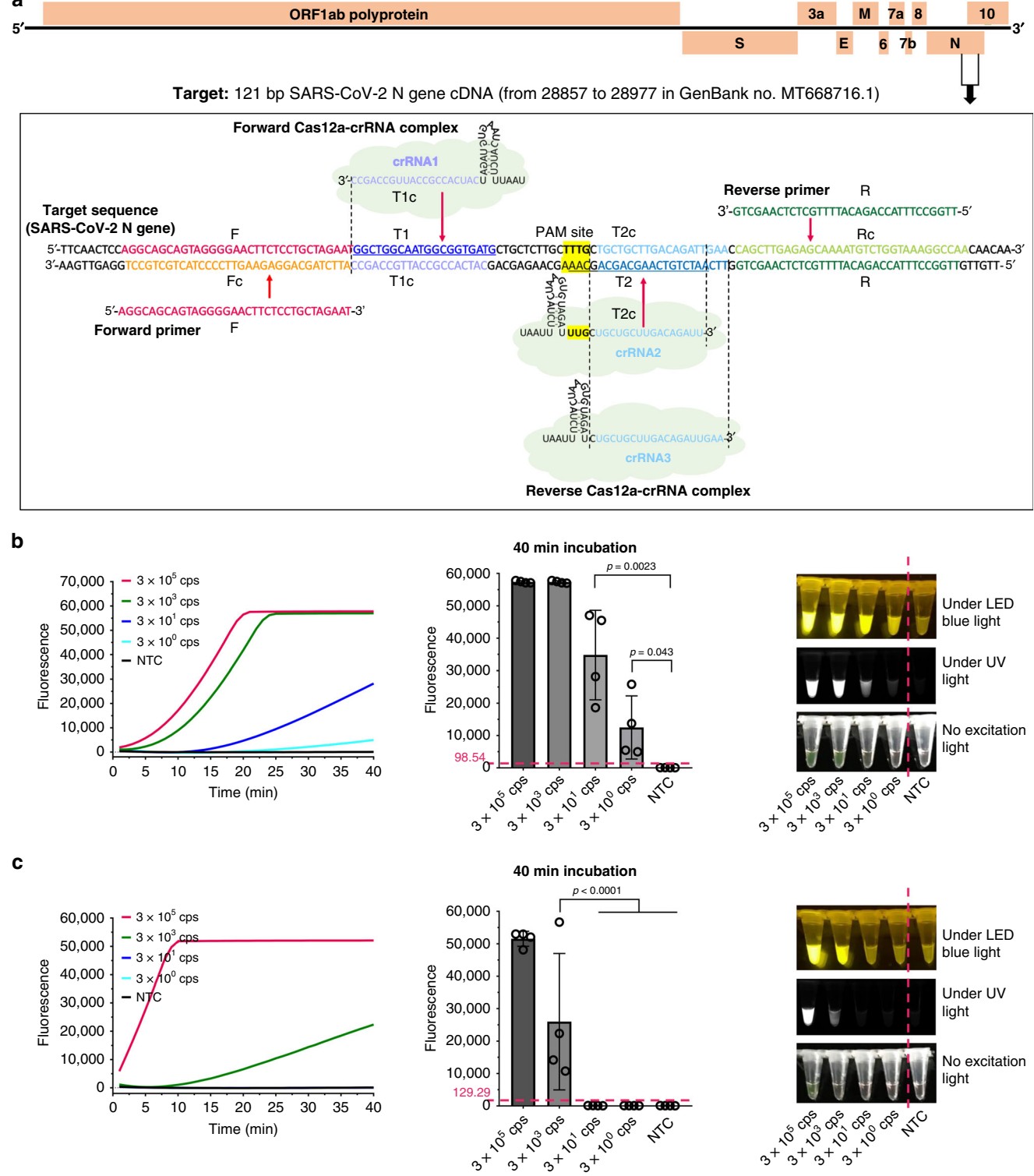

**Fig. 2 Primer design and the sensitivity of AIOD-CRISPR for the detection of the tenfold serial dilution of plasmid DNA containing SARS-CoV-2 N gene sequence (N plasmid). a** SARS-CoV-2 genome map with the detailed sequence information of the designed primers and crRNAs targeting the N gene sequence. For comparison purpose, the designed crRNA1 and crRNA2 were not limited by the PAM site (5′-TTTG-3′), while the crRNA3 was limited by the PAM site. **b** Real-time/endpoint AIOD-CRISPR assay using dual crRNAs (crRNA1&2) without PAM site limitation. Source data are provided as a Source Data file. **c** Real-time/endpoint AIOD-CRISPR assay using single crRNA (crRNA3) with PAM site limitation. Four replicates were run ($n = 4$). Source data are provided as a Source Data file. The horizontal dashed line indicates the cutoff fluorescence that was defined by the average intensity of NTC plus three times of the standard deviation. NTC, non-template control reaction. Error bars represent the means ± standard deviation (s.d.) from replicates. The unpaired two-tailed $t$-test was used to analyze the statistical significance. Source data is available for **b**, **c**.

conditions of ssDNA-FQ reporter (8 μM), primer concentration (0.32 μM), and ratio of crRNAs to Cas12a (1:1), we first evaluated the AIOD-CRISPR assays using single crRNA (crRNA1, crRNA2, and crRNA3) for the detection of $3 \times 10^6$ copies plasmid DNA. As shown in Supplementary Fig. 7b, the single crRNA3 with PAM sequence limitation showed the fastest fluorescence response than that of crRNA1 and crRNA2 without PAM sequence limitation. Next, we compared the performance of dual crRNAs (crRNA1&2 and crRNA1&3). As shown in Supplementary Fig. 8b, although the crRNA1&3 showed faster fluorescence response than the crRNA1&2, it triggered the nonspecific fluorescence signals in negative controls, which may potentially lead to an increased risk of false positive. Thus, we further determined the sensitivities of the AIOD-CRISPR assays with dual crRNA1&2 without PAM sequence limitation and single crRNA3 with PAM sequence limitation. As shown in Fig. 2b, c and Supplementary Fig. 9, the AIOD-CRISPR assay with crRNA1&2 without PAM sequence limitation can consistently detect three copies of SARS-CoV-2 plasmid DNA, which is much higher than that of single crRNA3 with PAM sequence. The lower sensitivity of single crRNA3 with PAM sequence in our AIOD-CRISPR assay is likely attributed to that the templates at low concentrations are rapidly cleaved by highly activated Cas12a from the crRNA3 with PAM sequence, thereby decreasing the amplification efficiency. Therefore, dual crRNAs without PAM sequence enables highly sensitive AIOD-CRISPR assay and eliminates the requirement of PAM sequence limitation.

**SARS-CoV-2 detection by AIOD-CRISPR assay.** To evaluate the detection specificity, we tested our AIOD-CRISPR assay using commercially available control plasmids containing the complete N gene from SARS-CoV-2 (SARS-CoV-2 _PC, Catalog # 10006625, IDT), SARS (SARS-CoV control, Catalog # 10006624, IDT), and Middle East respiratory syndrome (MERS; MERS-CoV (Middle East respiratory syndrome coronavirus) control, Catalog # 10006623, IDT), as well as the Hs_RPP30 control (Hs_RPP30_PC, Catalog # 10006626, IDT) with a portion of human RPP30 gene. Figure 3a and Supplementary Fig. 10 showed that only the reaction with SARS-CoV-2_PC had the positive signal in both real-time and visual detections, demonstrating that our developed AIOD-CRISPR assay possesses high specificity without cross reactions for non-SARS-CoV-2 targets.

Next, we used T7 promotor-tagged PCR and T7 RNA polymerase to prepare SARS-CoV-2 N gene RNA sequences (Supplementary Fig. 11) and developed reverse transcription AIOD-CRISPR (RT-AIOD-CRISPR) assay by supplementing the optimized avian myeloblastosis virus (AMV) reverse transcriptase (8 U; Supplementary Fig. 12). As shown in Fig. 3b and Supplementary Fig. 13, the RT-AIOD-CRISPR assay could consistently detect down to ~5 copies of SARS-CoV-2 N RNA targets in both real-time and visual detections.

Therefore, by targeting the SARS-CoV-2 N gene, our AIOD-CRISPR assay method was able to detect the nucleic acids with a sensitivity of few copies, providing a rapid, highly sensitive and specific method for SARS-CoV-2 detection.

**Clinical validation and instrument-free POC diagnostics.** Given the outstanding performance, the RT-AIOD-CRISPR assay was further applied to detect SARS-CoV-2 from COVID-19 patient samples. A total of 28 de-identified clinical swab samples (including eight COVID-19 positive samples) were tested in our RT-AIOD-CRISPR assay by using their RNA extracts. To ensure the detection reliability, each sample was tested twice in two independent assays. As shown in Fig. 4a and Supplementary Fig. 14, all eight COVID-19 positive samples were identified to be

SARS-CoV-2-positive by our real-time RT-AIOD-CRISPR assay in 40 min, which was also confirmed by endpoint visual detection (Fig. 4b and Supplementary Fig. 15). Further, our RT-AIOD-CRISPR assay results were consistent with those of CDC-approved RT-PCR method (Supplementary Table 1).

To further demonstrate its POC diagnostic application, we used a low-cost hand warmer (~$0.3 per bag) as the incubator of our RT-AIOD-CRISPR assay and detect COVID-19 patient samples. As shown in Fig. 5a, the AIOD-CRISPR assay tubes were directly placed on an air-activated hand warmer without need for any electric incubator. The endpoint fluorescence result can be observed by the naked eye under LED light. Figure 5b shows that two SARS-CoV-2-positive samples incubated in the hand warmer bag were visually detected and identified within as short as 20 min which excludes the nucleic acid extraction time. The longer the incubation time, the stronger the fluorescence signal of the positive samples. In addition, a similar result was achieved through analyzing the green value of the fluorescence images using the ImageJ software (Fig. 5c). Therefore, our AIOD-CRISPR method provides a simple, rapid, and visual approach for SARS-CoV-2 detection and has the potential to develop an instrument-free POC diagnostics of the COVID-19.

## Discussion

The emergence of the new coronavirus SARS-CoV-2, and its rapid spread through many countries, has been labeled as a global health emergency by the WHO[22]. Early diagnosis of these severe infections is crucial to prevent the rapid spread of this deadly virus globally. Nucleic acid amplification testing (e.g., PCR/RT-PCR) represents the most sensitive and specific method for the early detection of the pathogens[23,24], but current PCR technology is not suitable for rapid POC diagnostic application due to the need for specialized laboratory equipment and trained technicians. The limitations of current detection technology represent serious barriers for the real-time monitoring and detection of the highly contagious pathogens to prevent them spreading from person-to-person. Thus, there is an urgent need for a simple, easy-to-use, and inexpensive diagnostic approach.

In this study, we described a simple, rapid, ultrasensitive, and highly specific AIOD-CRISPR assay for the detection of SARS-CoV-2. This AIOD-CRISPR assay method is, to the best of our knowledge, the first system that allows all components to be incubated in one pot for CRISPR-based nucleic acid detection, enabling simple, all-in-one molecular diagnostics without need for separate and complex manual operations. Our AIOD-CRISPR assay takes advantage of dual crRNAs without PAM sequence limitation to enable highly sensitive, specific, and robust SARS-CoV-2 detection. Importantly, the detection results of the AIOD-CRISPR assay can be directly visualized by the naked eye, significantly simplifying the detection process and eliminating the need for separate lateral flow-based detection[18].

Compared to previously reported CRISPR-based nucleic acid detection methods[13,15,17,18,25–27], our versatile and robust AIOD-CRISPR assay has some distinctive advantages and provides a true single reaction system. In our AIOD-CRISPR assay, the components for both isothermal amplification and CRISPR-based detection are prepared in one-pot, completely circumventing the separate pre-amplification of target nucleic acids[13], or physical separation of Cas enzyme[27]. The AIOD-CRISPR assay enables rapid, ultrasensitive (few copies), and highly specific nucleic acid detection. We attribute rapid detection speed and ultrahigh sensitivity of our AIOD-CRISPR assay to: (i) the introduction of unique dual CRISPR-Cas12a detection methodology, (ii) the increased concentration of ssDNA-FQ reporters, and (iii) the

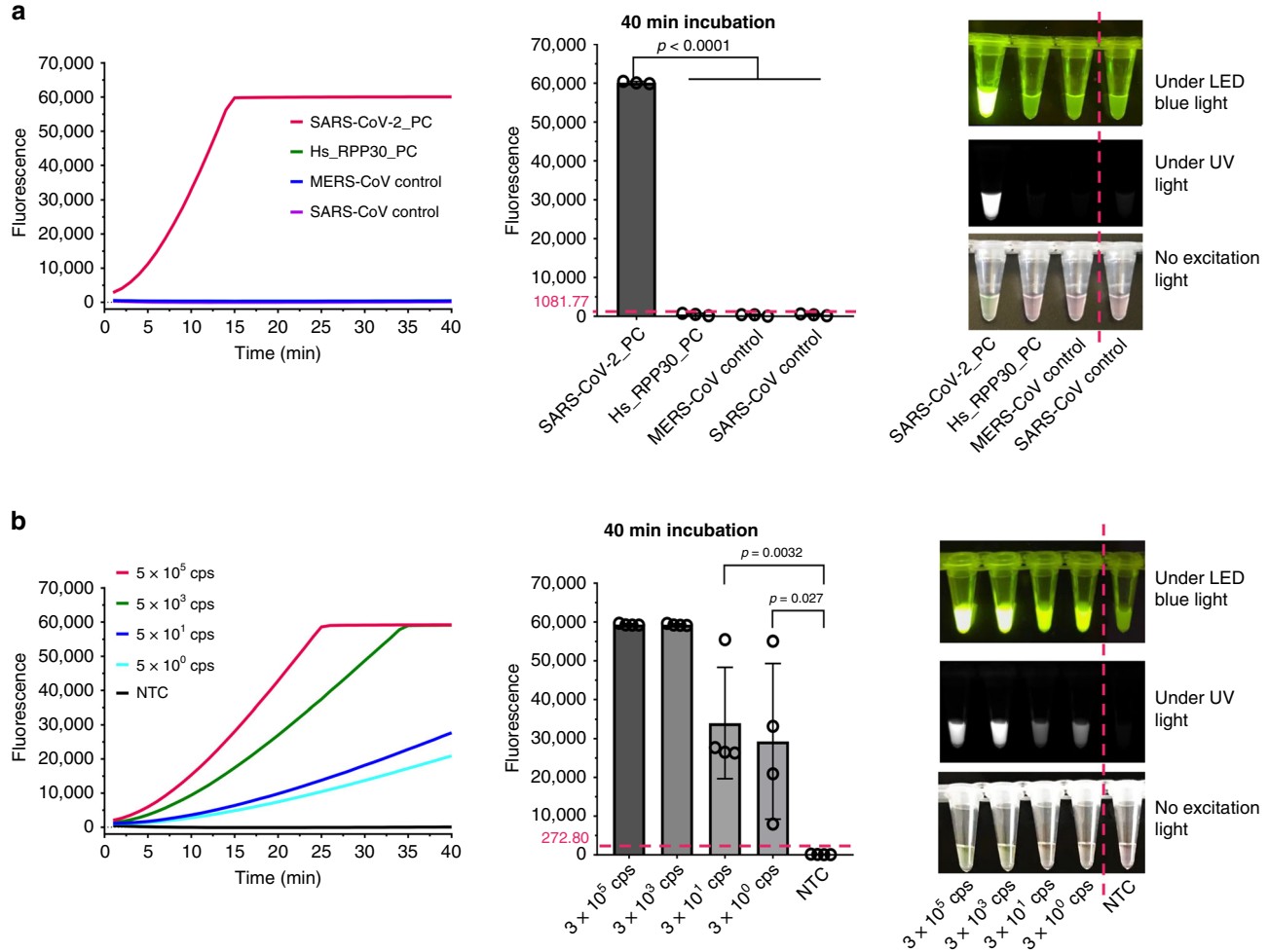

**Fig. 3 Specificity of AIOD-CRISPR and sensitivity of RT-AIOD-CRISPR for detection of synthetic SARS-CoV-2 N RNA sequences. a** Specificity of real-time/endpoint AIOD-CRISPR assay for SARS-CoV-2 N detection. Three replicates were run ($n = 3$). Source data are provided as a Source Data file. **b** Real-time/endpoint RT-AIOD-CRISPR detection of the tenfold serial dilution of synthetic SARS-CoV-2 N RNA sequences. Four replicates were run ($n = 4$). Source data are provided as a Source Data file. The horizontal dashed line indicates the cutoff fluorescence that was defined by the average intensity of NTC plus three times of the standard deviation. NTC, non-template control reaction. Error bars represent the means ± s.d. from replicates. The unpaired two-tailed $t$-test was used to analyze the statistical significance. Source data is available for **a**, **b**.

combination of RPA amplification and CRISPR-Cas12a-based detection. Also, the AIOD-CRISPR assay showed a high specificity in the SARS-CoV-2 detection without any cross-interaction with other sequences (e.g., SARS-CoV, MERS-CoV; Fig. 3a), which may be due to the nature of the CRIPSR-Cas12a's single-base specificity[28,29].

Although in this study, we demonstrated only qualitative detection of nucleic acids, we anticipate that our method is able to achieve semiquantitative detection by quantifying endpoint fluorescence intensities. By adding AMV reverse transcriptase, the AIOD-CRISPR assay can be easily developed as one-step RT-AIOD-CRISPR assay to detect RNA targets such as SARS-CoV-2 RNA, which facilitates the CRISPR-Cas12a-based RNA detection without need for the separate preparation of cDNA. To evaluate the validity and clinical applications of our AIOD-CRISPR assay, we adapted it to detect viral RNAs extracted from SARS-CoV-2 virus in nasal swab samples, achieving consistent detection results with that of RT-PCR method. Most importantly, we successfully demonstrated an instrument-free AIOD-CRISPR assay for SARS-CoV-2 detection in clinical samples by using a simple hand warmer. The cost of our AIOD-CRISPR assay is estimated ~$6 and can be significantly decreased when scaled-up for bulk production.

Further improvement and development are to integrate our AIOD-CRISPR assay into a disposable microfluidics chip platform[30–33], enabling fully integrated, sample to result, and multiplexed detection. On one hand, all reagents of the AIOD-CRISPR assay can be lyophilized and prestored in a disposable microfluidic chamber[34], which eliminates need for cold chains and enables rapid detection outside of a laboratory setting. On the other hand, multiplexing detection can be developed by combining multiplexed microfluidics technology[35–37]. Symptom of COVID-19 is nonspecific and similar to other respiratory illnesses[38]. Therefore, to enable effective disease treatment and management, it is critical to simultaneously detect and differentiate SARS-CoV-2 and other viral infections (e.g., influenza A/B, respiratory syncytial virus) by microfluidic-based multiplexed detection with single sample.

Since our AIOD-CRISPR assay generates strong fluorescence signals at the endpoint, it is possible to record, analyze, and report the detection results by taking advantage of ubiquitous smartphone technology[39–41]. The smartphone can be programmed to take fluorescence photos, convert the images into fluorescence intensity, analyze the data, and report the qualitative/semiquantitative test results. Further, the test results can be wirelessly transmitted to a website or remote server[40] and made available

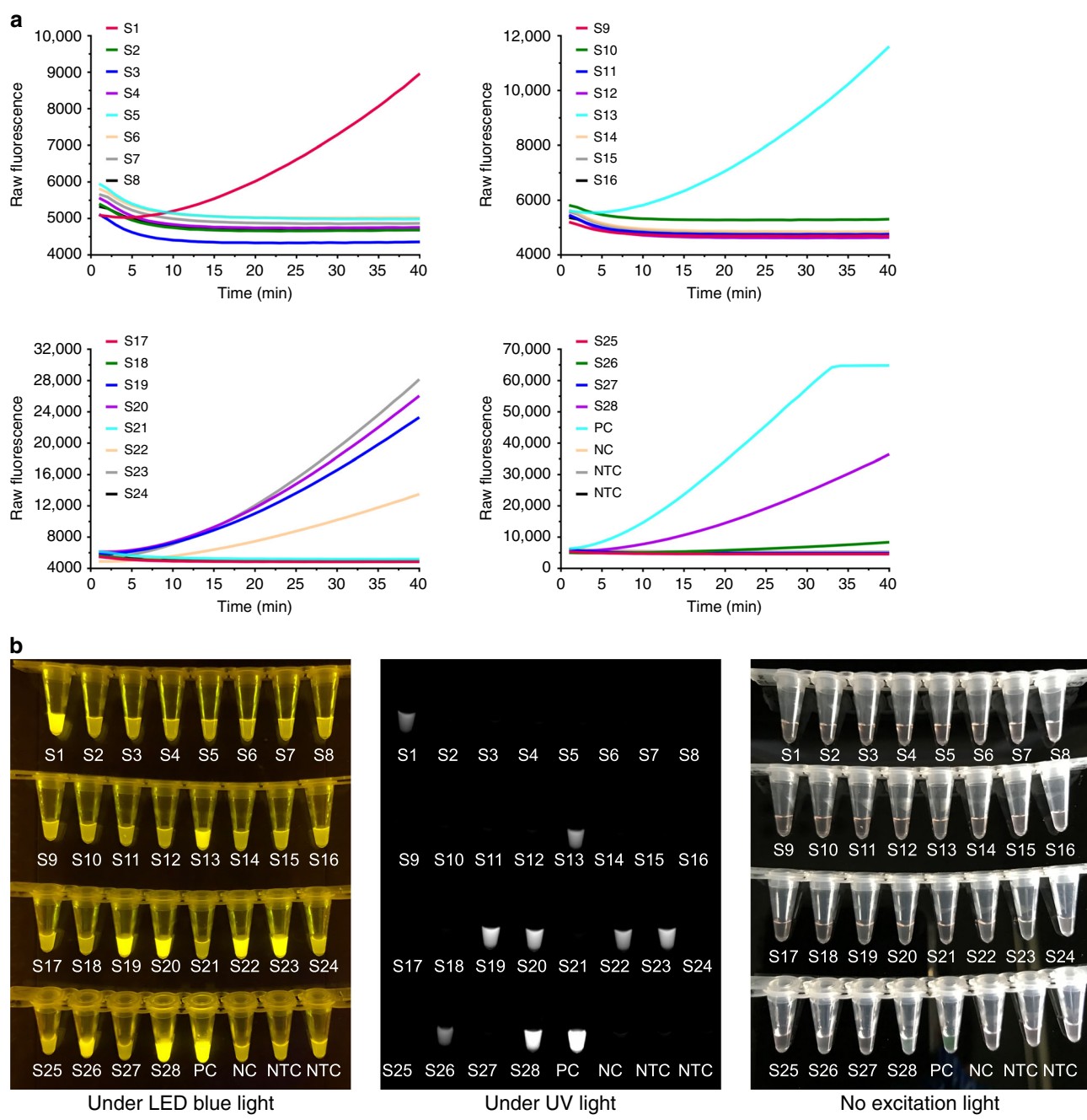

**Fig. 4 Detection of SARS-CoV-2 in clinical swab samples by RT-AIOD-CRISPR assay. a** Real-time RT-AIOD-CRISPR detection. **b** Endpoint fluorescence/visual detection after 40 min incubation. PC, positive controls with $4.6 \times 10^4$ copies of synthetic SARS-CoV-2 N RNA. S1-S28, clinical samples 1-28. NC, SARS-CoV-2-negative control reaction. NTC non-template control reaction.

together with GPS coordinates to the patient's doctor and public health officials. This is critical to allow simple, rapid, smart, connected disease diagnostics, and tracking.

In summary, the AIOD-CRISPR assay has been demonstrated to be a rapid, all-in-one, isothermal approach for nucleic acid (DNA and RNA) detection with ultrahigh sensitivity and single-base specificity. In turn, such simple and robust method has great potential in the future development of a next-generation POC molecular diagnostics technology for the rapid detection of infectious diseases (e.g., COVID-19) at home or in small clinics.

## Methods

**Materials**. Gel Loading Buffer II (Denaturing PAGE), PureLink Quick Gel Extraction Kit, and TURBO DNA-free Kit were purchased from Thermo Fisher

Scientific (Waltham, MA). EvaGreen dye (20×) was purchased from Biotium (Fremont, CA). TEMED, $(NH_4)_2S_2O_8$, 30% acrylamide/bis-acrylamide solution, 10× TBE Buffer, and SsoAdvanced Universal SYBR Green PCR Supermix were purchased from Bio-Rad Laboratories (Hercules, CA). EnGen Lba Cas12a (Cpf1) (100 μM), deoxynucleotide (dNTP) mix (10 mM of each), and AMV reverse transcriptase (10,000 U/mL) were purchased from New England BioLabs (Ipswich, MA). RNeasy MinElute Cleanup Kit was purchased from QIAGEN (Frederick, MD). RiboMAX Large Scale RNA Production Systems-T7 was purchased from Sigma-Aldrich (St. Louis, MO). TwistAmp Liquid Basic Kit was purchased from TwistDx Limited (Maidenhead, UK). The LED blue light illuminator (Maestrogen UltraSlim) was purchased from Fisher Scientific (Pittsburgh, PA). In all, 28 de-identified clinical swab samples (including eight COVID-19 positive samples) were tested and their viral RNAs were extracted by utilizing 140 μL of each sample and eluting with 140 μL of buffer of the QIAamp DSP Viral RNA Mini Kit (QIAGEN N.V., Venlo, The Netherlands). These samples were screened for SARS-CoV-2 by CDC-approved RT-PCR assays (Thermo Fisher Scientific Inc., Waltham, MA) prior to our AIOD-CRISPR assays. All clinical samples were de-identified and in

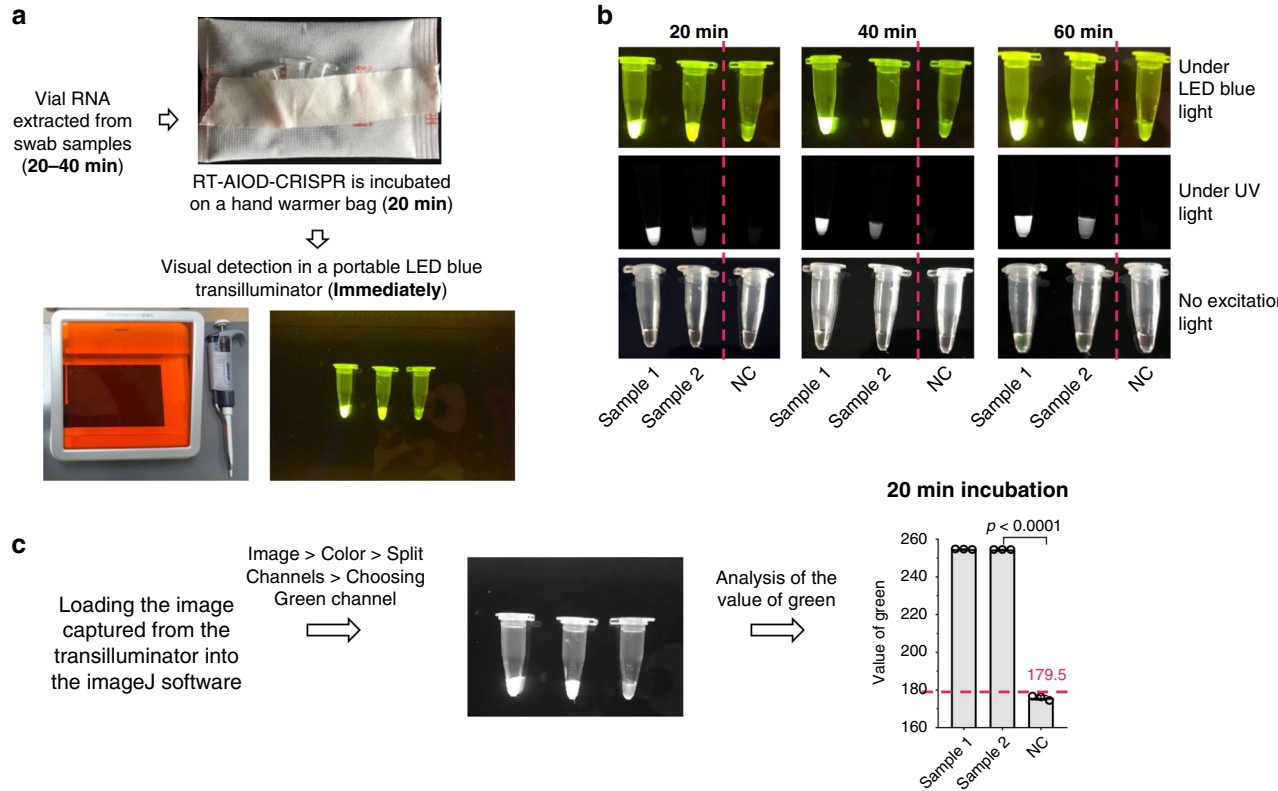

**Fig. 5 Instrument-free COVID-19 diagnostics by RT-AIOD-CRISPR assay. a** Workflow of the instrument-free POC diagnostics. **b** Visual detection after 20, 40, and 60-min incubation on the hand warmer bag. **c** Analysis of the green value for the fluorescence image (20-min incubation) using the ImageJ software. The horizontal dashed line indicates the cutoff fluorescence that was defined by the average intensity of NTC plus three times of the standard deviation. Source data are provided as a Source Data file. NC, the SARS-CoV-2-negative sample. Each measuring was run with three replicates ($n = 3$). Error bars represent the means ± s.d. from replicates. The unpaired two-tailed $t$-test was used to analyze the statistical significance. Source data is available for **c**.

compliance with ethical regulations and the approval of Institutional Review Board of the University of Health Center (protocol #: P61067).

**Design of primers and crRNAs**. The SARS-CoV-2 target sequence was 121 bp N gene fragment with the location from 28857 to 28977 in the SARS-CoV-2's genome (GenBank accession MT688716.1). Four sites of this target sequence were used to design the primers and crRNAs of the AIOD-CRISPR assay. These sites were checked to be highly conserved by using the GISAID's data on multiple sequence alignments analysis of 4663 SARS-CoV-2 genomes (as of August 14, 2020)[21]. Primer design for RPA and PCR was achieved by using the publicly available tools of OligoAnalyzer and Realtime PCR (https://www.idtdna.com/pages), respectively. The principle of designing RPA primers referred to the TwistAmp Assay Design Manual (https://www.twistdx.co.uk/en/support/manuals/twistamp-manuals). The crRNA with 20–24 nt size was also designed using the OligoAnalyzer and selected by against the MERS-CoV and SARS-CoV N genes. Oligonucleotides (primers and crRNAs), ssDNA-FQ reporters, the pUCIDT (Amp) plasmid with the 316 bp SARS-CoV-2 N gene sequence (from 28766 to 29081 in GenBank accession LC528233.1), and the control plasmids containing the complete N gene from SARS-CoV-2, SARS-CoV, and MERS-CoV, as well as the Hs_RPP30 control were synthesized from Integrated DNA Technologies (Coralville, IA). The sequence information of all used primers and crRNAs as well as the target inserted into a plasmid has been listed in the Supplementary Table 2.

**AIOD-CRISPR assays**. A step-by-step protocol describing the AIOD-CRISPR assay can be found at Protocol Exchange[42]. The AIOD-CRISPR system was prepared separately as Component A, B, and C. Component A contained 1× Reaction Buffer, 1× Basic E-mix, 14 mM MgOAc, 320 nM each of primers, and 1.2 mM dNTPs. Component B consisted of 4 or 8 μM of ssDNA-FQ reporters and 1× Core Reaction Buffer. Component C was the Cas12a-crRNA mix with 0.64 μM each of crRNAs and 1.28 μM EnGen Lba Cas12a. The concentration in each component was calculated based on the finally assembled 25-μL AIOD-CRISPR system. In a typical AIOD-CRISPR assay, 1 μL of the target solution was mixed with 20 μL of Component A and 2.5 μL of Component B. This assembled mixture was then mixed with 1.5 μL of Component C to form final 25 μL of AIOD-CRISPR system. For RT-AIOD-CRISPR assays, most components were the same as those in the AIOD-CRISPR system above,

except supplementing 0.32 U/μL AMV Reverse Transcriptase in Component A. Real-time fluorescence detection was carried out in the Bio-Rad CFX96 Touch Real-Time PCR Detection System. Visual detection was accomplished through imaging the tubes in the LED blue light illuminator or the Bio-Rad ChemiDoc MP Imaging System with its built-in UV channel. For visual detection based on the reaction solution's color change, 8 μM of ssDNA-FQ reporters should be used. All the reactions were incubated at 37 °C for 40 min or the denoted time in figures. The endpoint fluorescence was the raw fluorescence determined by the Real-Time PCR Detection System. A saturated fluorescence intensity was the maximum intensity which the Real-Time PCR Detection System could determine. After reaction, the AIOD-CRISPR solution was mixed with isometric Gel Loading Buffer II prior to 15% denaturing PAGE with 8 M urea and gel imaging in the Imaging System. Uncropped PAGE image is shown in Supplementary Fig. 16.

**In vitro RNA preparation using T7 RNA polymerase**. For SARS-CoV-2 N RNA preparation, the PCR system contained 1× SsoAdvanced Universal SYBR Green PCR Supermix, 0.4 μM of the forward T7 promotor-tagged primers, 0.4 μM of the reverse primers, and 1.0 μL of $1.3 \times 10^5$ copies/μL SARS-CoV-2 N plasmid solution. The thermal cycling was 2.5 min at 98 °C for initial denaturation, 35 cycles of 15 s at 95 °C for denaturation and 30 s at 60 °C for annealing and extension. The products of PCR were confirmed by agarose gel electrophoresis and Sanger sequencing. Afterwards, the products with the accurate sizes were extracted and purified using the Gel Extraction Kit. In vitro transcription was achieved through incubating the reaction system containing 8 μL of 5× T7 Transcription Buffer, 3 μL each of 100 mM rNTPs, 4 μL of the Enzyme Mix with T7 RNA polymerase, and 16 μL of the gel-extracted PCR products at 37 °C for 4 h. Then, the transcription products were treated by DNase (from the TURBO DNA-free TM Kit) to degrading the DNA and the RNA was extracted and purified using the RNeasy MinElute Cleanup Kit. The purity and concentration of the collected nucleic acid were determined using NanoDrop One/One Microvolume UV–Vis Spectrophotometry (Thermo Fisher Scientific).

**Statistics and reproducibility**. Statistical significances were analyzed by using the Prism 8 (GraphPad Software, version 8.0.1). The data involving endpoint fluorescence and threshold time to positive were all displayed with error bars which

represent mean ± s.d. from three or more than three replicates. The unpaired two-tailed *t*-test was applied to investigate the differences between groups and the threshold for defining significance was based on the *p* value <0.05. Unless otherwise specified, each image for visual detection shown in the corresponding figure is a representative of at least two independent experiments.

**Reporting summary**. Further information on research design is available in the Nature Research Reporting Summary linked to this article.

## Data availability

The authors declare that the data supporting the findings of this study are available within the paper and its supplementary information files or from the corresponding author upon reasonable request. Sequence information used in this study was from National Center for Biotechnology Information (NCBI) with GenBank accessions of MT688716.1 and LC528233.1 (https://www.ncbi.nlm.nih.gov/). Multiple sequence alignments analysis of 4663 SARS-CoV-2 genomes (as of August 14, 2020) are available from GISAID (https://www.gisaid.org/epiflu-applications/next-hcov-19-app/).The source data involving endpoint fluorescence, threshold time to positive, and the number of mutations in Fig. 2b, c, 3a, b, and 5c, and Supplementary Figs. 3b, c, 4b, c, 5b, c, 6a–e, 7c, 8c, and 12b, c, are provided as a Source Data file of this paper. Source data are provided with this paper.

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

## Acknowledgements

The work was supported, in part, by R01EB023607, R01CA214072, and R21TW010625.

## Author contributions

X.D. and C.L. conceived the technique, performed experiments, analyzed the data, and drafted the paper. K.Y. and Z.L. contributed to data collection and data review. R.V.L., E.B., and M.M.S. contributed to clinical samples, scientific advice, and resource for this study, and editing of the paper. C.L. supervised the whole project. All authors reviewed and approved the paper.

## Competing interests

The authors declare no competing interests.
