## [Peer Review File · Nature Communications]

Reviewers' Comments:

Reviewer #1:

Remarks to the Author:

This is a review of the manuscript by Ding, et al. titled "All-in-One Dual CRISPR-Cas12a (AIOD-CRISPR) Assay: A Case for Rapid, Ultrasensitive and Visual Detection of Novel Coronavirus SARS-CoV-2 and HIV virus at the Point of Care." The study is timely, as there is a global need for rapid and inexpensive detection assays for SARS-CoV-2. The study is also highly innovative, as the investigators have devised a strategy to detect viruses using the CRISPR technology, which is highly specific and easy to design.

The detection assays for SARS-CoV-2 and HIV viruses appear to exhibit high levels of sensitivity, with the ability to detect fewer than 10 viral RNA copies in a sample. The assays are rapid, with detection times generally occurring within 20-40 minutes. The investigators have engineered an instrument-free COVID-19 diagnostic assay, which is portable and could be used in non-laboratory settings. Altogether, these demonstrations made this study compelling with high potential impact.

Despite the compelling nature of the study, there were several flaws with the approach and the data figures. Many of these are substantial. These concerns are listed below:

1. The title and theme of the study emphasizes that the AIOD-CRISPR assay can be used for SARS-CoV-2 and HIV. This implies that these two viruses would be tested simultaneously. However, SARS-CoV-2 virus is not present in plasma, whereas HIV-1 virus is not present in nasopharyngeal or oropharyngeal specimens. Thus, the purpose of combining detection assays for these two very different viruses is unclear, since they would not be present in the same specimen.
2. Page 2, last paragraph, "Since the SARS-CoV-2 and HIV-1 are retroviruses": SARS-CoV-2 is a coronavirus, not a retrovirus.
3. Page 2, last line, "The AIOD-CRISPR method was clinically validated using COVID-19 patient samples." The method can only be "clinically validated" in a formal clinical trial overseen by the FDA or a similar governing agency. Testing the system with human samples does not mean that it has been "clinically validated." Please remove this claim.
4. Figure 1: This experiment shows the conditions required for a positive test: target sequence, dual crRNAs, Cas12a, ssDNA-FQ, and RPM mix. However, there is no testing for false negatives. The requirement of the binding of four complementary ~20 nt sequences (2 primers and 2 crRNAs) for a positive detection seems overly stringent and would likely lead to false negatives with the mutation of the SARS-CoV-2 genome at any of those four sites. This is particularly true since the four sequences do not contain degenerate nucleotides. Standard qRT-PCR detection assays for HIV-1 and SARS-CoV-2 only require the binding of two complementary primers, and the primers typically contain degenerate nucleotide sequences at positions with moderate or high sequence variability. In the proposed study, the likelihood for a false negative test would be relatively high, due to the requirement of four distinct binding events that each require ~20 nt of conserved sequences.
5. Figure 2: As noted in Comment #4, it seems that there would be a high risk of false negative results if the HIV-1 target sequences have nucleotide variation at the 2 primer binding sites or the 2 crRNA target sites. This experiment illustrates that concern. The authors used HIV-1 sequence GenBank No. EU770743.1, which is a common reference sequence and laboratory strain. A simple sequence alignment on www.hiv.lanl.gov reveals that the 32-nt forward primer is conserved in 455/9547 (4.8%) sequences, the 32-nt reverse primer is conserved in 92/9547 (1.0%) sequences, the 24-nt crRNA region is conserved in 3620/9547 (38.0%) sequences, and the 20-nt crRNA region is conserved in 324/9547 (3.4%) sequences (see table below). This assay has not been optimized to detect the global variants of HIV-1 sequences and would thus not be useful clinically.

F_HIV(EU770743.1):_CAAGCAGCCATGCAAATGTTAAAAGAAACCATC

F_HIV(K03455):____CAAGCAGCCATGCAAATGTTAAAAGAGACCATC (31/32)

F_HIV(AF324493):___CAAGCAGCCATGCAAATGTTAAAAGAGACCATC (31/32)

F_HIV(AB221005):__CAAGCAGCCATGCAAATGTAAAAGAACCCATC (31/32)
100% complementary: 455/9547 HIV-1/SIVcpz sequences

R_HIV(EU770743.1):_GTAGTTCCTGCTATGTCACCTCCCTTGGATC
R_HIV(K03455):____GTAGTTCCTGCTATGTCACCTCCCTTGGTTC (31/32)
R_HIV(AF324493):__GTAGTTCCTGCTATGTCACCTCCCTTGGTTC (31/32)
R_HIV(AB221005):__GTAGTTCCTGCTATGTCACCTCCCTTGGATC (32/32)
100% complementary: 92/9547 HIV-1/SIVcpz sequences

crRNA1_HIV(EU770743.1):_ATCCCATCTGCAGCTTCCTCATT
crRNA1_HIV(K03455):____ATCCCATCTGCAGCTTCCTCATT (24/24)
crRNA1_HIV(AF324493):__ATCCCATCTGCAGCTTCCTCATT (24/24)
crRNA1_HIV(AB221005):__ATCCCATCTGCAGCTTCCTCATT (22/24)
100% complementary: 3620/9547 HIV-1/SIVcpz sequences

crRNA2_HIV(EU770743.1):_TTGCACCAGGCCAGATAAGA
crRNA2_HIV(K03455):____TTGCACCAGGCCAGATGAGA (19/20)
crRNA2_HIV(AF324493):__TTGCACCAGGCCAGATGAGA (19/20)
crRNA2_HIV(AB221005):__TTGCACCAGGCCAGATAAGA (20/20)
100% complementary: 324/9547 HIV-1/SIVcpz sequences

6. Figure 3B: It is difficult distinguish the different conditions in the real-time fluorescence measurements because of the similarly colored red curves. Nevertheless, the following statement over-interprets the data: "the RT-AIOD-CRISPR assay was able to consistently detect 11 copies of HIV-1 RNA targets in both real-time and endpoint fluorescence visual detections." For the endpoint measurements, N=1, so the term "consistently" is misplaced.

7. Figure 3C: In this experiment, HIV-1 RNA from human plasma is detected by the RT-AIOD-CRISPR system. Are the HIV-1 samples in this experiment from a single HIV-infected patient? What was the plasma viral load of the specimen collected from the patient? In addition to the AIOD-CRISPR system, what other conventional or clinical assays were used to measure HIV-1 levels in these samples? How did the AIOD-CRISPR measurements compare to those other assays?

8. Figure 4 and p. 7: As discussed in Comments #4 and #5, the experiment shows that there is minimization for false positives, but it does not test for false negatives, due to sequence variation, limit of detection, or source of specimen. The sequence diversity of SARS-CoV-2 is much less than for HIV-1, but sequence variability should nevertheless be considered in the design of the primers and crRNAs. The authors assume the SARS-CoV-2 reference genome as GenBank ID LC528233, but the targeted region with contains sequence with variability that would likely lead to false negatives. For instance, the 33-nt forward primer contains two mismatches relative to SARS-CoV-2 GenBank ID MT412175. This is one of many other examples that suggest that the designed assay would not be useful clinically. I strongly suggest using a public database (like <https://www.gisaid.org/> or <https://www.ncbi.nlm.nih.gov/genbank/sars-cov-2-seqs/>) to optimize the design for primers and crRNA sequences and to utilize degenerate nucleotides whenever possible.

F_primer_LC528233: AGGCAGCAGTAGGGGAAGTTCTCCTGCTAGAAT
F_primer_MT412175: AGGCAGCAGTAGGGGAATTTCTCCTGGTAGAAT (31/33)

9. Discussion section, p. 10: "The emergence of the new coronavirus SARS-CoV-2, and its rapid spread through many countries, has been labeled as a global health emergency by the WHO. HIV is another deadly retrovirus..." Again, the authors have erred by stating that SARS-CoV-2 is a retrovirus. HIV is a lentivirus, which is part of the retrovirus family in the Revtraviricetes class of the Pararnavirae kingdom of viruses. The coronavirus family is in the Pisoniviricetes class of the Orthornavirae kingdom of viruses. These viruses are very different.

Reviewer #2:

Remarks to the Author:

The manuscript by Ding et al describes a new method for virus detection. The method is based on the combination of two known techniques: recombinase polymerase amplification (RPA) and CRISPR-Cas enzymes. The RPA have already been used for detection, and the authors here combine RPA with a CRISPR-Cas system with the purpose of adding two levels of signal amplification through DNA replication and collateral cleavage activities to improve the detection limit. This combination is novel and has not been reported for SARS-CoV-2 detection.

The approach of visual coronavirus SARS-CoV- 2 and HIV-1 virus detection method based on the dual CRISPR-Cas12a assay, is clever, but there are some limitations that might prevent it from becoming widely adopted.

1. The assay is not as convenient as the authors claimed, DNA/RNA isolation, multiple enzymes/proteins, primer design and sophisticated system optimization are required, holding inherent limitations in simplicity and usability.
2. A large variation exists during 3 times measurements in Figure 2D, the reproducibility of this assay may have a problem. "False negative" or "False positive" may exist in clinical use.
3. For practical application, the authors collected 8 samples for virus infection diagnosis. The sample size is too small and the results are not representative. I hope the author can collect more samples to verify the feasibility of the method. The authors should provide the information of patient samples.
4. Cost of the assay: One of the main motivations of the authors is to offer a sensitive and affordable virus detection assay. It is true that the developed assay overcomes the need for antibodies. However, this assay uses different types of enzymes: polymerase, recombinase and CRISPR-Cas12a, which would significantly increase the cost of the assay. It would be helpful if the authors could estimate the cost of the assay for a single test.
5. 'The detection region of the RT-AIOD-CRISPR was verified by Sanger sequencing (Figure 7B)', should be corrected as Figure S7B.
6. It may be better to explain the principle of RT-AIOD-CRISPR assay by a schematic, in order to distinguish it from AIOD-CRISPR assay.

Reviewer #3:

Manuscript Review

Manuscript by Dr Liu et al., "All-in-One Dual CRISPR-Cas12a (AIOD-CRISPR) Assay: A Case for Rapid, Ultrasensitive and Visual Detection of Novel Coronavirus SARS-CoV-2 and HIV virus at the Point of Care".

The manuscript by Liu et al. introduces a new detection assay using CRISPR-Cas mechanism. The main claim of this manuscript is that using two (dual) crRNA enhances the sensitivity of CRISPR-based detection assays. However, this conclusion is not supported by the data, and other minor conclusions in the manuscript require additional experiments to be confirmed.

- 1) This claim is mainly discussed in Figure 2b and c and the last paragraph on page 4:
 - First, the authors should add standard deviations to be able to comment on the significance of any change in the data.
 - The differences in RFU between crRNA2, crRNA2+crRNA1, and 2*crRNA2 is NOT significant, and likely falls within the standard deviation.
 - The only significant difference in this figure is between crRNA1 and crRNA2, which suggests that crRNA2 is much more specific to the target than crRNA 1.
 - The authors also conclude that doubling the concentration of crRNA1 or crRNA2 does not have any effect on RFU. This conclusion cannot be drawn from this figure as it needs an additional experiment to confirm that the concentrations used for crRNA1 and crRNA2 are not already saturating the reaction medium, in which case doubling the concentration will only create a competitive reaction for the same sites and will not result in RFU increase. The authors should study the change in RFA intensity as a function of the crRNA concentrations, similarly to what they did with the ssDNA-FQ and the primer concentrations.
 - Figure 2c: the images of the tubes showing 3 fluorescent tubes for crRNA2 and 4 fluorescent tubes for crRNA1+crRNA2 is misleading. According to Figure 2b, crRNA2 can sometimes generate higher RFU than crRNA1+crRNA2, which means that crRNA2 alone can also generate 4 fluorescent tubes. This is where standard deviation is an important element.

- 2) Page 6: the authors claim that: "Although incubated for 40 min, the AIOD-CRISPR assay could detect and identify 1.2 copies of HIV-1 DNA in just 1-min incubation...".
 - According to figure 3A and B, the RFU values are too low and too close to reliably differentiate between various concentrations after 1 min. A reliable detection time seems to be between 15 and 30 min incubation. In addition to the lack of standard deviations, this experiment lacks another important data: the background fluorescence. There is always a baseline of background fluorescence in any fluorescence-based assay or in any assay involving biological samples due to autofluorescence, and the absence of such information in Figure 3A and 3B makes it difficult to claim the detection limit.

- 3) The authors claim a detection time of 1 min. We have already mentioned that the RPA reaction and fluorescence detection requires 15-30 min. The authors should clearly state that this time does not include RNA extraction from clinical samples which requires at least 30 min (using lysis, washing and elution buffers and spin columns). The total detection time for the technique proposed in this manuscript would be around 1 hour in the best-case scenario.

- 4) I am don't how researchers can prepare a solution with 1 copy in a volume of 25 microliters. At some point during dilution, it's hard to know in which aliquot you "1 copy" is. I am always skeptical when researchers claim the detection of a single molecule/single copy/single bacterium, because the probability of having the target in your sample decreases exponentially with increasing number of dilutions. Any dilution below 10 units will be extremely unreliable unless you analyze all the diluted samples.

Given that the main claim of the authors is not supported by the data provided in the manuscript, I recommend rejection.

Response to Reviewer 1's Comments:

We are grateful to the reviewer 1 for his/her time, enthusiasm, and comments. Below, we reproduce the reviewer 1's comment in italic and our response in blue regular print.

Reviewer #1 (Remarks to the Author):

This is a review of the manuscript by Ding, et al. titled "All-in-One Dual CRISPR-Cas12a (AIOD-CRISPR) Assay: A Case for Rapid, Ultrasensitive and Visual Detection of Novel Coronavirus SARS-CoV-2 and HIV virus at the Point of Care." The study is timely, as there is a global need for rapid and inexpensive detection assays for SARS-CoV-2. The study is also highly innovative, as the investigators have devised a strategy to detect viruses using the CRISPR technology, which is highly specific and easy to design.

The detection assays for SARS-CoV-2 and HIV viruses appear to exhibit high levels of sensitivity, with the ability to detect fewer than 10 viral RNA copies in a sample. The assays are rapid, with detection times generally occurring within 20-40 minutes. The investigators have engineered an instrument-free COVID-19 diagnostic assay, which is portable and could be used in non-laboratory settings. Altogether, these demonstrations made this study compelling with high potential impact.

Response: Thank you for your positive comments!

Despite the compelling nature of the study, there were several flaws with the approach and the data figures. Many of these are substantial. These concerns are listed below:

1. The title and theme of the study emphasizes that the AIOD-CRISPR assay can be used for SARS-CoV-2 and HIV. This implies that these two viruses would be tested simultaneously. However, SARS-CoV-2 virus is not present in plasma, whereas HIV-1 virus is not present in nasopharyngeal or oropharyngeal specimens. Thus, the purpose of combining detection assays for these two very different viruses is unclear, since they would not be present in the same specimen.

Response: In this manuscript, we detected SARS-CoV-2 and HIV viruses in different samples, respectively, to demonstrate the versatility and robustness of our AIOD-CRISPR assay. In its title, we included the words "a Case" to indicate that it is just an application example to detect SARS-CoV-2 and HIV viruses. To make it clearer, we have rephrased some sentences in the revised manuscript.

2. Page 2, last paragraph, "Since the SARS-CoV-2 and HIV-1 are retroviruses": SARS-CoV-2 is a coronavirus, not a retrovirus.

Response: Thanks! It is corrected.

3. Page 2, last line, "The AIOD-CRISPR method was clinically validated using COVID-19 patient samples." The method can only be "clinically validated" in a formal clinical trial overseen by the FDA or a similar governing agency. Testing the system with human samples does not mean that it has been "clinically validated." Please remove this claim.

Response: Thanks! Done.

4. Figure 1: This experiment shows the conditions required for a positive test: target sequence, dual crRNAs, Cas12a, ssDNA-FQ, and RPM mix. However, there is no testing for false negatives. The requirement of the binding of four complementary ~20 nt sequences (2 primers and 2 crRNAs) for a positive detection seems overly stringent and would likely lead to false negatives with the mutation of the SARS-CoV-2 genome at any of those four sites. This is particularly true since the four sequences do not contain degenerate nucleotides. Standard qRT-PCR detection assays for HIV-1 and SARS-CoV-2 only require the binding of two complementary primers, and the primers typically contain degenerate nucleotide sequences at positions with moderate or high sequence variability. In the proposed study, the likelihood for a false negative test would be relatively high, due to the requirement of four distinct binding events that each require ~20 nt of conserved sequences.

Response: We prepared one combined response to comments 4, 5 and 8. Please see it in the comment 8.

5. Figure 2: As noted in Comment #4, it seems that there would be a high risk of false negative results if the HIV-1 target sequences have nucleotide variation at the 2 primer binding sites or the 2 crRNA target sites. This experiment illustrates that concern. The authors used HIV-1 sequence GenBank No. EU770743.1, which is a common reference sequence and laboratory strain. A simple sequence alignment on www.hiv.lanl.gov reveals that the 32-nt forward primer is conserved in 455/9547 (4.8%) sequences, the 32-nt reverse primer is conserved in 92/9547 (1.0%) sequences, the 24-nt crRNA region is conserved in 3620/9547 (38.0%) sequences, and the 20-nt crRNA region is conserved in 324/9547 (3.4%) sequences (see table below). This assay has not been optimized to detect the global variants of HIV-1 sequences and would thus not be useful clinically.

F_HIV(EU770743.1):_ CAAGCAGCCATGCAAATGTTAAAAGAAACCATC

F_HIV(K03455):_____ CAAGCAGCCATGCAAATGTTAAAAGAGACCATC (31/32)

F_HIV(AF324493):__ CAAGCAGCCATGCAAATGTTAAAAGAGACCATC (31/32)

F_HIV(AB221005):__ CAAGCAGCCATGCAAATGTTAAAAGAACCCATC (31/32)

100% complementary: 455/9547 HIV-1/SIVcpz sequences

R_HIV(EU770743.1):_ GTAGTTCCTGCTATGTCACTTCCCCTTGGATC

R_HIV(K03455):_____ GTAGTTCCTGCTATGTCACTTCCCCTTGGTTC (31/32)

R_HIV(AF324493):__ GTAGTTCCTGCTATGTCACTTCCCCTTGGTTC (31/32)

R_HIV(AB221005):__ GTAGTTCCTGCTATGTCACTTCCCCTTGGATC (32/32)

100% complementary: 92/9547 HIV-1/SIVcpz sequences

crRNA1_HIV(EU770743.1):_ ATCCCATTCTGCAGCTTCCTCATT

crRNA1_HIV(K03455):_____ ATCCCATTCTGCAGCTTCCTCATT (24/24)

crRNA1_HIV(AF324493):__ ATCCCATTCTGCAGCTTCCTCATT (24/24)

crRNA1_HIV(AB221005):__ ATCCCATCTTGCAGCTTCCTCATT (22/24)

100% complementary: 3620/9547 HIV-1/SIVcpz sequences

crRNA2_HIV(EU770743.1):_TTGCACCAGGCCAGATAAGA

crRNA2_HIV(K03455):_____TTGCACCAGGCCAGATGAGA (19/20)

crRNA2_HIV(AF324493):__TTGCACCAGGCCAGATGAGA (19/20)

crRNA2_HIV(AB221005):__TTGCACCAGGCCAGATAAGA (20/20)

100% complementary: 324/9547 HIV-1/SIVcpz sequences

Response: We prepared one combined response to comments 4, 5 and 8. Please see it in the comment 8.

6. *Figure 3B: It is difficult distinguish the different conditions in the real-time fluorescence measurements because of the similarly colored red curves. Nevertheless, the following statement over-interprets the data: "the RT-AIOD-CRISPR assay was able to consistently detect 11 copies of HIV-1 RNA targets in both real-time and endpoint fluorescence visual detections." For the endpoint measurements, N=1, so the term "consistently" is misplaced.*

Response: We are sorry that we do not clearly distinguish the different real-time fluorescence curves. To make it clearer, we have changed the colours of different real-time fluorescence curves.

For both real-time fluorescence detection and endpoint measurement, three replicates were run (n=3). To make it clearer, all results of independent assays were shown in **Figure 3A, 3B, S6 and S8**.

7. *Figure 3C: In this experiment, HIV-1 RNA from human plasma is detected by the RT-AIOD-CRISPR system. Are the HIV-1 samples in this experiment from a single HIV-infected patient? What was the plasma viral load of the specimen collected from the patient? In addition to the AIOD-CRISPR system, what other conventional or clinical assays were used to measure HIV-1 levels in these samples? How did the AIOD-CRISPR measurements compare to those other assays?*

Response: The AcroMetrix™ HIV-1 plasma control samples were purchased from Thermo Fisher Scientific (Waltham, MA)". We compared the performance of our AIOD-RISPR assay with that of the RT-PCR method that is considered as "gold standard" for HIV virus detection, such as HIV viral load. As shown in **Figure 3C and S9**, our AIOD-CRISPR assay showed comparable sensitivity with RT-PCR method.

8. *Figure 4 and p. 7: As discussed in Comments #4 and #5, the experiment shows that there is minimization for false positives, but it does not test for false negatives, due to sequence variation, limit of detection, or source of specimen. The sequence diversity of SARS-CoV-2 is much less than for HIV-1, but sequence variability should nevertheless be considered in the design of the primers and crRNAs. The authors assume the SARS-CoV-2 reference genome as GenBank ID LC528233, but the targeted region with contains sequence with variability that would likely lead to false negatives. For instance, the 33-nt forward primer contains two mismatches relative to SARS-CoV-2 GenBank ID MT412175. This is one of many other examples that suggest that the designed assay would not be useful clinically. I strongly*

suggest using a public database (like <https://www.gisaid.org/> or <https://www.ncbi.nlm.nih.gov/genbank/sars-cov-2-seqs/>) to optimize the design for primers and crRNA sequences and to utilize degenerate nucleotides whenever possible.

F_primer_LC528233: AGGCAGCAGTAGGGGAACTTCTCCTGCTAGAAT

F_primer_MT412175: AGGCAGCAGTAGGGGAATTTCTCCTGGTAGAAT (31/33)

Response to Comments 4, 5 and 8: Thank you for your constructive comments! In this study, we mainly develop a new general CRISPR-based nucleic acid detection method (AIOD-CRISPR) for simple, rapid and inexpensive point of care molecular diagnostics of infectious diseases. SARS-CoV-2 and HIV-1 are just used as application examples.

Previous literatures showed that RPA isothermal amplification has toleration for targets with sequence variation (Daher et al., *Clinical chemistry*, 2016, 62(7): 947-58; Lobato et al., *Trac Trends in analytical chemistry*, 2018, 98: 19-35). Thus, degenerate nucleotides are not introduced for RPA isothermal amplification, including two *Science* papers that firstly reported CRISPR-based nucleic acid detection: i) DETECTR method (Chen et al., *Science*, 2018, 360(6387): 436-9) and ii) SHERLOCK method (Gootenberg et al., *Science*, 2017, 356(6336): 438-42).

In addition, even for the CDC-approved RT-PCR assay (<https://wwwnc.cdc.gov/eid/article/26/8/20-1246-t1>) for SARS-CoV-2 detection, their primers do not contain degenerate nucleotides, which is consistent with the recently reported CRISPR-Cas12a based SARS-CoV-2 detection method (Broughton et al., *Nature Biotechnology*, 2020, 16:1-5).

Lastly but not least, we have tested clinical samples with our AIOD-CRISPR assay and obtained comparable results with RT-PCR method, confirming its clinical utility.

9. Discussion section, p. 10: “The emergence of the new coronavirus SARS-CoV-2, and its rapid spread through many countries, has been labeled as a global health emergency by the WHO. HIV is another deadly retrovirus...” Again, the authors have erred by stating that SARS-CoV-2 is a retrovirus. HIV is a lentivirus, which is part of the retrovirus family in the *Revtroviricetes* class of the *Paramnavirae* kingdom of viruses. The coronavirus family is in the *Pisoniviricetes* class of the *Orthonavirae* kingdom of viruses. These viruses are very different.

Response: Thanks again! We have corrected them.

Response to Reviewer 2's Comments:

We thank the reviewer 2 for his/her appreciation of the value of our work. Below, we reproduce the reviewer 2's comments in italic and our response in blue regular print.

Reviewer #2 (Remarks to the Author):

The manuscript by Ding et al describes a new method for virus detection. The method is based on the combination of two known techniques: recombinase polymerase amplification (RPA) and CRISPR-Cas enzymes. The RPA have already been used for detection, and the authors here combine RPA with a CRISPR-Cas system with the purpose of adding two levels of signal amplification through DNA replication and collateral cleavage activities to improve the detection limit. This combination is novel and has not been reported for SARS-CoV-2 detection.

The approach of visual coronavirus SARS-CoV- 2 and HIV-1 virus detection method based on the dual CRISPR-Cas12a assay, is clever, but there are some limitations that might prevent it from becoming widely adopted.

Response: Thank you for your positive comments!

1. The assay is not as convenient as the authors claimed, DNA/RNA isolation, multiple enzymes/proteins, primer design and sophisticated system optimization are required, holding inherent limitations in simplicity and usability.

Response: Like any nucleic acid amplification testing (e.g., PCR/RT-PCR), it is necessary to include isolated DNA/RNA, enzymes/proteins, and primers to achieve highly sensitive and specific molecular detection. But unlike RT-PCR method ("gold standard" of SARS-CoV-2 detection), our AIOD-CRISPR assay does not require expensive real-time PCR machine (\$40K) and can visually detect SARS-CoV-2 by using a hand warmer (\$ 0.3) (**Figure 6**), which has significant potential for developing next-generation point-of-care molecular diagnostics.

2. A large variation exists during 3 times measurements in Figure 2D, the reproducibility of this assay may have a problem. "False negative" or "False positive" may exist in clinical use.

Response: We have added new experimental data, and included standard deviation and unpaired t-test results. As shown in **Figure 2C** and **S2**, the reproducibility of our assay is good and the background fluorescence of negative controls is very stable, which may contributed to highly specific fluorescence signal amplification of the CRISPR-Cas12a detection.

3. For practical application, the authors collected 8 samples for virus infection diagnosis. The sample size is too small and the results are not representative. I hope the author can collect more samples to verify the feasibility of the method. The authors should provide the information of patient samples.

Response: As per reviewer 2's suggestion, the information of patient samples were added in **Figure S15**. We are working with clinicians to collect and test more patient samples.

4. *Cost of the assay: One of the main motivations of the authors is to offer a sensitive and affordable virus detection assay. It is true that the developed assay overcomes the need for antibodies. However, this assay uses different types of enzymes: polymerase, recombinase and CRISPR-Cas12a, which would significantly increase the cost of the assay. It would be helpful if the authors could estimate the cost of the assay for a single test.*

Response: Done. The cost of our AIOD-CRISPR assay is estimated ~\$ 6 and can be significantly decreased when scaled-up for bulk production.

5. *'The detection region of the RT-AIOD-CRISPR was verified by Sanger sequencing (Figure 7B)', should be corrected as Figure S7B.*

Response: Thanks. Done.

6. *It may be better to explain the principle of RT-AIOD-CRISPR assay by a schematic, in order to distinguish it from AIOD-CRISPR assay.*

Response: Like conventional PCR and RT-PCR assay, the major difference between RT-AIOD-CRISPR and AIOD-CRISPR assay is that the RT-AIOD-CRISPR includes an additional reverse transcription (RT) step (RNA is reverse transcribed into complementary DNA (cDNA)) by adding reverse transcriptase because both SARS-CoV-2 and HIV are RNA viruses. We have added some description on it in the revised manuscript.

Response to Reviewer 3's Comments:

We are grateful to the reviewer 3 for his/her constructive comments and careful review, which help us further improve the manuscript. Below, we reproduce the reviewer 3's comments in italic and our response in blue regular print.

Reviewer #3 (Remarks to the Author):

Manuscript by Dr Liu et al., "All-in-One Dual CRISPR-Cas12a (AIOD-CRISPR) Assay: A Case for Rapid, Ultrasensitive and Visual Detection of Novel Coronavirus SARS-CoV-2 and HIV virus at the Point of Care".

The manuscript by Liu et al. introduces a new detection assay using CRISPR-Cas mechanism. The main claim of this manuscript is that using two (dual) crRNA enhances the sensitivity of CRISPR based detection assays. However, this conclusion is not supported by the data, and other minor conclusions in the manuscript require additional experiments to be confirmed.

Response: As reflected in the title of our manuscript, there are two main claims in our study: **i) one is "all-in-one" CRISPR-based nucleic acid detection.** As far as we know, it is the first time to enable RPA isothermal amplification and CRISPR detection in "one-pot" format, eliminating the need for separate pre-amplification, transferring of amplified products and lateral flow detection, and **ii) another is to develop dual crRNAs strategy to enhance the sensitivity of CRISPR based detection.** To further support this claim, we have added independent experimental data and performed data analysis (e.g., standard deviation, unpaired t-test). As shown in **Figure 2C, S2 and S3B**, the experimental data showed that introducing dual crRNAs can improve the detection sensitivity of our AIOD-CRISPR assays.

1) This claim is mainly discussed in Figure 2b and c and the last paragraph on page 4:

- First, the authors should add standard deviations to be able to comment on the significance of any change in the data.

Response: Thanks! Done. Please see **Figure 2B and 2C**.

*- The differences in RFU between crRNA2, crRNA2+ceRNA1, and 2*crRNA2 is NOT significant, and likely falls within the standard deviation.*

Response: To make it clearer, we have added the standard deviation and unpaired t-test results. As mentioned above, introducing dual crRNAs can improve the detection sensitivity of our AIOD-CRISPR assays as shown in **Figure 2C, S2 and S3B**.

- The only significant difference is in this figure is between crRNA1 and crRNA2, which suggests that crRNA2 is much more specific to the target than crRNA 1.

Response: This is just the reason why we compare the performance of our dual crRNAs (crRNA1 and crRNA2) with that of 2*crRNA2, not 2*crRNA1.

- The authors also conclude that doubling the concentration of crRNA1 or crRNA2 does not have any effect on RFU. This conclusion cannot be drawn from this figure as it needs an additional experiment to confirm that the concentrations used for crRNA1 and crRNA2 are not already saturating the reaction medium, in which case doubling the concentration will only create a competitive reaction for the same sites and will not result in RFU increase. The authors should study the change in RFA intensity as a function of the crRNA concentrations, similarly to what they did with the ssDNA-FQ and the primer concentrations.

Response: We have added experimental data and data analysis results (e.g., standard deviation, unpaired t-test) as shown in **Figure 2B**.

- Figure 2c: the images of the tubes showing 3 fluorescent tubes for crRNA2 and 4 fluorescent tubes for crRNA1+crRNA2 is misleading. According to Figure 2b, crRNA2 can sometimes generate higher RFU than crRNA1+crRNA2, which means that crRNA2 alone can also generate 4 fluorescent tubes. This is where standard deviation is an important element.

Response: To make it clearer, we have included standard deviation, cut-off fluorescence curves and added independent experimental data as shown in **Figure 2C, S2 and S3B**. As mentioned above, these experimental data showed that introducing dual crRNAs improves the detection sensitivity of our AIOD-CRISPR assays.

2) Page 6: the authors claim that: "Although incubated for 40 min, the AIOD-CRISPR assay could detect and identify 1.2 copies of HIV-1 DNA in just 1-min incubation..."

- According to figure 3A and B, the RFU values are too low and too close to reliably differentiate between various concentrations after 1 min. A reliable detection time seems to be between 15 and 30 min incubation. In addition to the lack of standard deviations, this experiment lacks another important data: the background fluorescence.

Response: Since one-min testing is not our major claim in this study, we have deleted the claims that "Although incubated for 40 min, the AIOD-CRISPR assay could detect and identify 1.2 copies of HIV-1 DNA in just 1-min incubation..."

We are sorry that the background fluorescence baselines of previous **Figure 3A and 3B** are not clear because we used the minimum Y value of 5,000 which results in overlap between X-axis and baselines. To make them clearer, we have redrawn these figures with the minimum Y value of 0. Please see **Figure 3A and 3B** in the revised manuscript.

As per reviewer 3's suggestion, we have added standard deviations and background cut-off intensity curves. The background cut-off intensity was calculated as the mean plus three times the standard deviation.

There is always a baseline of background fluorescence in any fluorescence-based assay or in any assay involving biological samples due to autofluorescence, and the absence of such information in Figure 3A and 3B makes it difficult to claim the detection limit.

Response: Again, we are sorry that the baselines of background fluorescence are not clear in previous **Figure 3A and 3B** because we used the minimum Y value of 5,000. To make them clearer, we have redrawn these figures with the minimum Y value of 0. Please see **Figure 3A and 3B** in the revised manuscript.

3) *The authors claim a detection time of 1 min. We have already mentioned that the RPA reaction and fluorescence detection requires 15-30 min. The authors should clearly state that this time does not include RNA extraction from clinical samples which requires at least 30 min (using lysis, washing and elution buffers and spin columns). The total detection time for the technique proposed in this manuscript would be around 1 hour in the best-case scenario.*

Response: In the website of the TwistDx™ Limited (<https://www.twistdx.co.uk/en/rpa>), the vendor of RPA reagents, it clearly mentions that “**Results are typically generated within 3-10 minutes.**” It should be pointed out that that they use conventional fluoresce probe, not CRISPR-Cas12a, to detect the fluorescence signal of RPA isothermal amplification. In our AIOD-CRISPR assay, we take advantage of **signal amplification effect of CRISPR-Cas12a** to detect the fluorescence signal of RPA isothermal amplification in “**one-pot**” format. Thus, it is not surprising that our AIOD-CRISPR assay may detect the nucleic acid within 1 min as we demonstrated in **Figure S1**. As mentioned above, since one-min testing is not our major claim in this study, we have deleted this claim.

As per reviewer 3’s suggestion, we have added the nucleic acid extraction step in **Figure 6A** and stated that sample RNA extraction is not included in our AIOD-CRISPR assay.

4) *I am don’t how researchers can prepare a solution with 1 copy in a volume of 25 microliters. At some point during dilution, it’s hard to know in which aliquot you “1 copy “ is. I am always skeptical when researchers claim the detection of a single molecule/single copy/single bacterium, because the probability of having the target in your sample decreases exponentially with increasing number of dilutions. Any dilution below 10 units will be extremely unreliable unless you analyze all the diluted samples. Given that the main claim of the authors is not supported by the data provided in the manuscript, I recommend rejection.*

Response: The DNA or RNA samples with different concentration are prepared by 10-fold dilution of the nucleic acid sample solution with known concentration. The 10-fold dilution method has been widely used to prepare serial dilutions to determine analytical sensitivity (or limit of detection (LOD)) during nucleic acids analysis as a routine method. For instance, in recent *Nature Biotechnology* paper on CRISPR-based SARS-CoV-2 detection (Broughton et al., *Nature Biotechnology*, 2020, 16:1-5), the analytical sensitivity of CDC qPCR was estimated to be 1 copy of SARS-CoV-2 (**Figure 2d** of *Nature Biotechnology* paper) by the 10-fold dilution method.

To provide more convincing evidence, we have added the experimental results of independent replication assays. Our experimental data showed that we can detect 1.2 copies of HIV DNA (**Figure 3A**) and 1.3 copies SARS-CoV-2 DNA (**Figure 4B**). For HIV RNA and SARS-CoV-2 RNA detection, the sensitivity is, respectively, 11 copies (**Figure 3B**) and 4.6 copies (**Figure 4D**). For consistency, we have deleted the claim of “nearly single-molecule level sensitive detection” in the revised manuscript.

Given that the main claim of the authors is not supported by the data provided in the manuscript, I recommend rejection.

Reviewers' Comments:

Reviewer #1:

Remarks to the Author:

This is a review of the revised manuscript by Ding, et al. titled "All-in-One Dual CRISPR-Cas12a (AIOD-CRISPR) Assay: A Case for Rapid, Ultrasensitive and Visual Detection of SARS-CoV-2 Using All-in-One CRISPR Cas12a (AIOD-CRISPR) Assay." Although the original manuscript described a system to detect both HIV and SARS-CoV-2, the revised manuscript focuses entirely on SARS-CoV-2. This eliminates many of my concerns with the study and greatly improves the clarity and focus of the manuscript. In fact, the authors have addressed all of my concerns, with one exception, as described below.

In my previous review, I noted that the system might be susceptible to false negatives, particularly since it required the binding of four distinct complementary ~20 sites. The authors have provided sufficient evidence from literature of the robustness of the RPA isothermal amplification system. They have also tested the system with 28 clinical samples to validate its robustness. However, due to the constant evolution of the SARS-CoV-2 genome, it is necessary to acknowledge the possibility of the emergence of viral variants that might not be efficiently detected by this system. Using the public databases that I previously mentioned (<https://www.gisaid.org/> or <https://www.ncbi.nlm.nih.gov/genbank/sars-cov-2-seqs/>), the authors should note whether any currently circulating viral strains contain mutations in any of the targeted regions, and they should report the frequencies of such strains. This is a straightforward exercise that might help assessing the risk of the emergence of undetectable strains and the potential for false negative testing results.

Reviewer #2:

Remarks to the Author:

The authors have satisfactorily addressed reviewer comments.

Following are some suggestions for the improvement of the manuscript:

1. Practical application of assay: It would be better to specify the total time taken to test patient samples using this method, including the time sample pretreatment (eg. DNA/RNA isolation) and fluorescence testing.
2. It should be '#5' not '#4' in the second paragraph of AIOD-CRISPR assay system of Results.

Reviewer #3:

None

REVIEWERS' COMMENTS:

Reviewer #1 (Remarks to the Author):

This is a review of the revised manuscript by Ding, et al. titled "All-in-One Dual CRISPR-Cas12a (AIOD-CRISPR) Assay: A Case for Rapid, Ultrasensitive and Visual Detection of SARS-CoV-2 Using All-in-One CRISPR Cas12a (AIOD-CRISPR) Assay." Although the original manuscript described a system to detect both HIV and SARS-CoV-2, the revised manuscript focuses entirely on SARS-CoV-2. This eliminates many of my concerns with the study and greatly improves the clarity and focus of the manuscript. In fact, the authors have addressed all of my concerns, with one exception, as described below.

In my previous review, I noted that the system might be susceptible to false negatives, particularly since it required the binding of four distinct complementary ~20 sites. The authors have provided sufficient evidence from literature of the robustness of the RPA isothermal amplification system. They have also tested the system with 28 clinical samples to validate its robustness. However, due to the constant evolution of the SARS-CoV-2 genome, it is necessary to acknowledge the possibility of the emergence of viral variants that might not be efficiently detected by this system. Using the public databases that I previously mentioned (<https://www.gisaid.org/> or <https://www.ncbi.nlm.nih.gov/genbank/sars-cov-2-seqs/>), the authors should note whether any currently circulating viral strains contain mutations in any of the targeted regions, and they should report the frequencies of such strains. This is a straightforward exercise that might help assessing the risk of the emergence of undetectable strains and the potential for false negative testing results.

Response: As per reviewer 1's suggestion, we have searched the databases from GISAID and analyzed the mutation rate (Lines 150-153, page 4 in the main manuscript, and Supplementary Fig. 6 in the Supplementary Information).

Reviewer #2 (Remarks to the Author):

1. Practical application of assay: It would be better to specify the total time taken to test patient samples using this method, including the time sample pretreatment (eg. DNA/RNA isolation) and fluorescence testing.

Response: We have provided the specific time in Figure 5a. Also, we specified the detection time in the uploaded step-by-step protocol in the Protocol Exchange.

2. It should be '#5' not '#4' in the second paragraph of AIOD-CRISPR assay system of Results.

Response: Thank you for pointing it out. It was corrected in our revised manuscript.